# Effects of Cabbage-Apple Juice Fermented by *Lactobacillus plantarum* EM on Lipid Profile Improvement and Obesity Amelioration in Rats

**DOI:** 10.3390/nu12041135

**Published:** 2020-04-18

**Authors:** Sihoon Park, Hee-Kyoung Son, Hae-Choon Chang, Jae-Joon Lee

**Affiliations:** Department of Food and Nutrition, Chosun University, Gwangju 61452, Korea; sihun6312@naver.com (S.P.); kyoung1033@naver.com (H.-K.S.); hcchang@chosun.ac.kr (H.-C.C.)

**Keywords:** *Lactobacillus plantarum* EM, cabbage-apple juice, high-fat diet, anti-obesity, hypolipidemic

## Abstract

This study aimed to investigate the potential of cabbage-apple juice, fermented by *Lactobacillus plantarum* EM isolated from kimchi, to protect against obesity and dyslipidemia that are induced by a high-fat diet in a rat model. Male rats were fed a modified AIN-93M high-fat diet (HFD), the same diet supplemented with non-fermented cabbage-apple juice, or the same diet supplemented with fermented cabbage-apple juice for eight weeks. In the HFD-fermented cabbage- apple juice administered groups the following parameters decreased: body weight, liver and white fat pad weights, serum triglyceride (TG), total cholesterol (TC), LDL-cholesterol, insulin, glucose and leptin levels, TG levels, while HDL-C and adiponectin levels in serum increased as compared with the HFD group. The HFD-fed rats that were supplemented with fermented cabbage-apple juice exhibited significantly lower fatty acid synthase (FAS), acetyl-CoA carboxylase (ACC), and malic enzyme gene expression levels when compared to the exclusively HFD-fed rats. The anti-obesity and hypolipidemic effects were marginally greater in the fermented juice administered group than in the non-fermented juice administered group. These results suggest that cabbage-apple juice—especially fermented cabbage-apple juice—might have beneficial effects on lipid metabolism dysfunction and obesity-related abnormalities. However, further studies are necessary for analyzing the biochemical regulatory mechanisms of fermented juice for obesity amelioration and lipid metabolic homeostasis.

## 1. Introduction

An increase in fruit and vegetable intake has been consistently reported to reduce mortality due to cardiovascular disease and the risk of hypertension and stroke [1,2,3]. Fruits and vegetables are rich in potassium, folic acid, vitamins, dietary fiber, and phenol compounds. These compounds support the homeostasis regulation by decreasing oxidative stress, enhancing blood lipid metabolism, reducing blood pressure, and increasing insulin resistance [4,5,6,7,8]. It is recommended a minimum intake amounting to 1/5 of a daily diet to achieve the health-promoting effects of fruits and vegetables.

The main components of apples—the fruit of the apple tree, *Malus domestica,* a species of deciduous trees of the family Rosaceae of the order Rosales that belongs to dicotyledonous plants—are sugars and organic acids that have organoleptic qualities. Among the sugar components, 11–12% consists of oligosaccharides. In addition, the fruit contains a rich content of carotenoids, dietary fibers, vitamins, minerals, and antioxidant substances. The content of phenolic compounds, including procyanin, hydroxycinnamic acid, and its derivatives, phloridzin, chlorogenic acid, caffeic acid, catechins, and epicatechins is especially high, contributing to the preventive effects of apples on cardiovascular disease, diabetes, hypertension, and cancer [9,10,11]. Apple intake is known to have an effect on body weight reduction, not only in humans, but also in experimental animals [12,13,14,15].

Cabbage (*Brassica oleraces* L.) is a vegetable of the family cruciferous with a long history of cultivation. It is known to be particularly rich in nutrients, such as lysine, linolenic acid, β-carotene, vitamin C, dietary fiber, lutein, and zeaxanthin, as well as glucosinolates—the bioactive substances known to prevent cancer, enhance immune function [16,17], and reduce cholesterol or lipid levels [18]. It is also enriched with natural polyphenol compounds, including caffeic acid, ferulic acid, ρ-coumaric acid, phenolic acids, flavonols, and anthocyanidins [19]. When compared to the other vegetables of the same family, cabbage contains high levels of S-methylmethionine (SMM), which has been reported to suppress the secretion of gastric juice while facilitating cellular regeneration in tumor tissues and promoting anti-inflammation, pain inhibition, and the prevention of lipid accumulation [20]. S-methyl-l-cysteine sulfoxide in cabbages has also been reported to have an effect on reducing serum cholesterol levels [21]. In addition, recent research has shown the possible preventive and protective effects of β-carotene on hepatic steatosis, liver damage, dyslipidemia, diet-induced obesity, oxidative stress, inflammation, and fibrosis [22,23].

Fermentation proceeds with the addition of sugars, yeast, or microorganisms, such as lactic acid bacteria (LAB) to raw material, leading to the activation of a diversity of enzymes in the raw material and the consequent production of various functional substances, while the nutrients contained in the raw material are converted to a more easily digestible and absorbable form [24]. Thus, numerous studies have focused on the fermentation of natural food ingredients and its use in the development of functional foods with health benefits. Recently, various studies have been conducted on the development and functionality of fruit or vegetable juices fermented with LAB that exhibit an abundance of diverse bioactive substances [9,25,26,27,28]. Among the LAB used in fermenting fruits and vegetables, *Lactobacillus plantarum* (*L. plantarum*) is one of the most common species used as a probiotic, which has been reported to reduce body fat in mice [29] and exert inhibitory effects on adipogenesis in 3T3-L1 cells [30]. Apple juice fermented with *L. plantarum* ATCC14917 has been shown to increase the cytoprotective effects against oxidative stress by enhancing the bioavailability of phenolic substances, in contrast to non-fermented apple juice [9], while carrot juice fermented with *L. plantarum* NCU116 has been shown to exert preventive effects on type 2 diabetes [26]. In addition, when compared to raw cabbage juice, lactic acid-fermented sauerkraut juice increases the activity and gene expression levels of antioxidant enzymes in the liver [31].

*L. plantarum* is generally recognized as being safe on the basis of the long history of human consumption of *Lactobacilli* in food. Among the LAB isolated from kimchi, *L. plantarum* EM exhibits excellent survival and adhesion in the gut without developing resistance to antibiotics, and shows high cholesterol revomal by growing, resting, and even dead cells based on the high cholesterol-binding capacity of its cell wall fraction [32]. The experimental animals fed a high-fat diet acquired diet-induced obesity with consequent visceral fat increase, dyslipidemia, hyperinsulinemia, and/or fatty liver—a phenomenon that is known to occur similarly in the human body [33]. This implies that a study using a rat model with high-fat diet-induced obesity to explore the prevention of obesity and hyperlipidemia, as well as the efficacy of therapeutic supplements, is likely to yield significant findings that can also be applied to the human body. Thus, this study used a rat model, in which obesity was induced by a diet with 45% of total kcal from a high-fat diet, with the aim to assess the anti-obesity and lipid metabolism-enhancing effects of cabbage-apple juice and compare the effects with or without fermentation by *L. plantarum* EM.

## 2. Materials and Methods

### 2.1. Preparation of Fermented Cabbage-Apple Juice

Cabbage (*Brassica oleracea* var. Capitata) and apple (*Malus pumila* var. *dulcissima* Koidz) were cleaned under running water, and then pressed, separately, while using a juice extractor (HD-RBF09; Hurom, Gimhae, Korea). The two juices were then blended together in equal volumes. When we tested effect of various combinations of cabbage and apple juice to sensory panels, 1:1 ratio showed the highest score (data not shown). Thus, we selected the 1:1 ratio in this study. *L. plantarum* EM cultured overnight in MRS broth was centrifuged at 10100× *g* for 15 min at 4 °C (Hanil Science, Incheon, Korea), resuspended in sterilized distilled water, and then inoculated (up to 10^7^ CFU/mL) into the juice. The resultant preparation was designated as fermented cabbage-apple (FCA) juice in this study. The juice without *L. plantarum* EM inoculum was designated as non-fermented cabbage-apple (CA) juice. The prepared juice was fermented at 15 °C for five days; thereafter, the juice was stored at 4 °C for 21 days. The FCA juice contained approximately 9.01~9.11 log CFU of *L. plantarum* EM/mL, and the strain was not found in the CA juice.

### 2.2. Sample Analyses

The pH values of the samples were determined using a pH meter (Denver, Arvada, CO, USA). The total acidity of the samples was analyzed by titrating the diluted sample with 100 mmol/L NaOH until pH 8.3. Sugar contents of the samples were investigated using a saccharimeter (Atago pocket PAL-3, Atago Co., Ltd, Tokyo, Japan). Protein, fat, ash, moisture, and diatary fiber contents were determined using A.O.A.C methods [34]. The organic acids contents were analyzed according to the method described by Sturm et al. [35] using high-performance liquid chromatography (HPLC; Thermo Scientific, Finnigan Spectra System, Waltham, MA, USA). The free sugar contents in the samples were determined using HPLC. The HPLC conditions described by Richmond et al. [36] were used with some modifications. Total polyphenol contents were identified by the Folin–Ciocalten method [37], while using tannic acid as a standard. The absorbance was read at 725 nm. All of the experiments were performed in triplicate. The content of total glucosinolates in samples was analyzed by HPLC (HPLC; Thermo Scientific, Finnigan Spectra System, Waltham, MA, USA) according to the method of ISO [38] with slight modification.

### 2.3. Animals and Experimental Design

The experimental animals consisted of 24 male, five-week old Sprague Dawley rats purchased from Central Lab. Animal, Inc. (Seoul, Korea). After a week of adaptation to solid formula feed (Research Diets, Inc., New Brunswick, NJ, USA) at the Lab Animal Center of Chosun University, the rats were divided among each test group based on a randomized block design, with eight rats allocated to each group; each rat was isolated and maintained in a stainless steel cage. The test groups were, as follows: (1) high-fat diet group (HFD); (2) high-fat diet and cabbage-apple juice administration group (HFD-CA group); and, (3) high-fat diet and fermented cabbage-apple juice administration group (HFD-FCA group). For the high-fat diet, the AIN-93G diet (D12451; Research Diets, Inc. New Brunswick, NJ, USA) was used to ensure that a fat content of 45% per calorie was supplied. For the HFD-CA and HFD-FCA groups, the respective juice was administrated daily by oral gavage in 10 mL/kg of body weight (Zonde needle, JD-S-124; Jeungdo B&P Co., Ltd., Seoul, Korea) and concurrently fed HFD for eight weeks. For the HFD group, the rats were administered with an equal volume of physiological saline instead of the juice. The lighting was controlled on a 12 h light/dark cycle (lights on from 08:00–20:00) and the temperature of the feeding room was maintained at 18 ± 2 °C. Body weight and food intake were measured once weekly at the same fixed time and the rate of body weight gain was calculated by subtracting the weight before the experiment from the final weight and then dividing it by the weight before the experiment. The food intake and water consumption were monitored daily. The Institutional Animal Care and Use Committee of Chosun University approved the animal experimental protocol used in this study (CIACUC2019-A0003).

### 2.4. Blood and Tissue Sample Processing

After the eight-week feeding regimen, the rats were fasted for 12 h after oral administration, and then sacrificed by decapitation. The collected blood was centrifuged at 1100× *g* at 4 °C for 15 min to isolate the serum for storage at −70 °C until subsequent analysis. The liver and white fat pads (i.e., epididymal, mesenteric, retroperitoneal, and perirenal fat pads) were immediately extracted after the blood collection and their weights were measured immediately. The tissue weight was calculated as a relative weight per 100 g post-fasting body weight prior to autopsy. The tissue samples were stored at −70 °C until subsequent analysis to measure the lipid content.

### 2.5. Serum Biomarkers and Hepatic and Adipose Tissue Lipids

Triglycerides (TGs), total cholesterol (TC), and HDL-cholesterol in the serum were measured while using a blood biochemical analyzer (Fuji Dri-Chem 3500, Fujifilm, Tokyo, Japan). An enzyme assay kit (Biovision Inc., Mountain View, CA, USA) was used to measure LDL/VLDL-cholesterol. For the lipid contents in the liver and the white fat pads, the lipids were extracted while using the Folch method [39], a portion of which were used to measure the TG and TC contents following the methods of Biggs et al. [40] and Zlatkis and Zak [41], respectively.

### 2.6. Serum Biochemical Parameters

Glucose content and the activities of alanine aminotransferase (ALT), aspartate aminotransferase (AST), alkaline phosphatase (ALP), and lactate dehydrogenase (LDH) in the serum, were measured using the blood biochemical analyzer (Fuji Dri-Chem 3500). The levels of leptin and adiponectin in the serum secreted by adipose tissue were measured while using a leptin mouse/rat enzyme immunoassay (EIA) kit (Quantikine & Immuno Assay kit, R&D Systems, Minneapolis, MN, USA) and the adiponectin rat EIA (ALPCO Diagnostics, Salem, NH, USA), respectively, based on a sandwich-type enzyme-linked immunosorbent assay (ELISA), and then analyzed at 450 nm using a plate reader (Spectra Max 250; Molecular Devices, San Jose, CA, USA). Serum insulin levels were measured using an insulin radioimmunoassay kit (Eiken Chemical Co., Ltd., Tokyo, Japan).

### 2.7. Hepatic RNA Extraction and Reverse Transcription-Polymerase Chain Reaction (RT-PCR) Analysis

An RNeasy^®^ Mini Kit (Qiagen, Hilden, Germany) was used to isolate RNA from the liver and reverse transcribed by using AccuPower RT Premix (BIONEER Corp., Daejeon, Korea), according to the manufacturer’s instructions. A RT-PCR analysis (TaKaRa Biochemicals, Tokyo, Japan) was performed using the forward primer F (5′-CAACGCCTTCACACCACCTT-3′) and reverse primer R (5′-AGCCCATTACTTCATCAAAGATCCT-3′) for acetyl-CoA carboxylase (ACC); F (5′-TGCTCCCAGCTGCAAG-3′) and R (5′-GTATCCTCGGGACCGGTTAT-3′) for fatty acid synthase (FAS); F (5′-CGACCAG-CAAAGCTGAGTGTT-3′) and R (5′-CTGCCGCTGGCAAAGATC-3′) malic enzyme (ME); F (5′-GTTTGGCAGCGGCAACTAA-3′) and R (5′- GGCATCACCCTGGTACAACTC-3′) for glucose 6-phosphate dehydrogenase (G6PDH); and, F 5′-GTGGGGCGCCCCAGGCACCAGGGC-3′ and R (5′-CTCCTTAATGTCACGCACGATTTC-3′ for β-actin. One microliter of oligo (dT) (Invitrogen/Thermo Fisher Scientific, Carlsbad, CA, USA) and DEPC were added to 1 μg of the isolated RNA to make a 20 μL mixture; this was placed in AccuPower^®^ RT-premix (Bioneer, Seoul, Korea) for cDNA synthesis with the following reaction conditions: 42 °C for 60 min and 94 °C for 5 min PCR conditions were as follows: 94 °C for 3 min, 30 s at 94 °C (denaturation); 30 s at 62 °C (annealing); 45 s at 72 °C (extension) × 30 cycles, 72 °C for 10 min, with maintenance at 4 °C thereafter. The PCR products were analyzed via 2% agarose gel electrophoresis in order to detect the expression of each gene and the house-keeping gene β-actin was used as the control for mRNA levels. The data were analyzed using the Alpha Ease FC software (Alpha Innotech Corporation, San Leandro, CA, USA).

### 2.8. Histopathological Analysis of Hepatic Tissue and Adipocytes in the Epididymal Adipose Tissue

The samples of liver tissue extracted immediately after the autopsy of rats were collected and fixed using 4% paraformaldehyde solution. Next, using the Cryocut Microtome (Leica CM1800; Wetzler, Germany) at −25 °C, 3–4 μm thick sections were prepared and then attached to the slide for drying. After Oil-Red O staining followed by the sequential steps of washing, neutralization, and dehydration, the slide was sealed with a mounting agent. The state of the tissue was then observed under the light microscope.

The epididymal fat pads of rats were cut to equal sizes, and then fixed using 10% formalin for 24 h. Under running water, any excess fixing agents were removed. The moisture in the tissue was removed using ethyl alcohol and the alcohol in the tissue was removed using xylene, after which the tissue space was filled through paraffin treatment. The slides were prepared using 5 μm microsections; after hematoxylin and eosin (H&E) staining, images of the adipocytes were captured using an optical microscope (TS100; Nikon, Tokyo, Japan), and the hepatic adipocytes from each test group were compared with respect to size, using an image analyzer program (National Institute of Mental Health, Bethesda, MD, USA).

### 2.9. Statistical Analysis

The experimental results were statistically analyzed while using the Statistical Package for Social Science (SPSS) program (SPSS Version 21.0, IBM Corp., Armonk, NY, USA) and each group was expressed as mean ± standard error. One-way analysis of variance was carried out to test the significance of the intergroup differences in average values; at *p* < 0.05, Tukey’s post-hoc test was performed.

## 3. Results

### 3.1. pH, Acidity, Nutrient Components, Organic Acid and Free Sugar Compositions, Total Polyphenol, and Total Glucosinolates Contents of Juice Samples

The results showing pH, acidity, proximate constituents, organic acid and free sugar compositions, and total polyphenol content of juice samples are listed in Table 1, respectively. The pH values of the CA juice and FCA juice were 4.08 and 3.63, respectively, showing that the FCA juice had the lowest pH. The acidity of the CA juice and FCA juice were 1.26 and 1.58%, respectively. The carbohydrate and fat contents were higher in CA juice than in FCA juice. However, the dietary fiber contents were higher in FCA juice than in CA juice. The contents of total organic acids were significantly higher in FCA juice than in CA juice. Acetic and lactic acid content was high in FCA juice, while citric acid and fumaric acid contents were higher in CA juice than in FCA juice. The contents of total free sugars were significantly lower in FCA juice than CA juice. The total polyphenol content of the FCA juice were slightly higher than that of the CA juice. However, there no significant differences in total glucosinolates content between CA juice and FCA juice.

### 3.2. Body Weight Gain and Food Intake

The changes in body weight and food intake of the rats fed a high-fat diet with orally administered cabbage-apple juice or fermented cabbage-apple juice are shown in Table 2. Body weight gains decreased significantly in the HFD-CA and HFD-FCA groups by 20.2% and 21.9%, respectively, as compared to the HFD group, but there was no significant difference between the juice-administrated groups. The changes in body weight during the eight-week period indicated weight gain every week in all test groups, with the HFD-CA and HFD-FCA groups displaying in changes body weight from week 4, resulting in significant differences from week 6 when compared to the HFD group (Figure 1). Food intake did not show significant differences among the test groups.

### 3.3. Liver and White Fat Pad Weights

Table 3 provides the weights of the liver, each white fat pad, and total white fat pads per 100 g body weight of the rats that were fed a high-fat diet with orally administered cabbage-apple juice or fermented cabbage-apple juice. The liver weights displayed a significant decrease in the HFD-CA and HFD-FCA groups by 13.7% and 16.3%, respectively, as compared to the HFD group. The juice-supplemented groups, HFD-CA and HFD-FCA, did not differ in liver weight. The total weight of white fat pads as the sum of the weights of epididymal, mesenteric retroperitoneal, and perirenal fat pads decreased in the HFD-CA and HFD-FCA groups by 6.8% and 15.7%, respectively, as compared to the HFD group. The weights of each white fat pad—epididymal, mesenteric, and retroperitoneal, composing the visceral fat—decreased in the HFD-CA and HFD-FCA groups when compared to the HFD group by approximately 10.1–15.1%, 9.8–17.6%, and 6.7–14.5%, respectively. However, there were no significant differences in the perirenal fat pads weight among the experimental groups. When compared to the HFD group, the group administered with fermented cabbage-apple juice (HFD-FCA) showed a significant reduction in the weights of epididymal, mesenteric, and retroperitoneal fat pads, whereas the group that was administered with cabbage-apple juice (HFD-CA) showed a significant reduction in the weight of epididymal fat pads only.

### 3.4. Biochemical Indicators of Hepatic Function

Figure 2 shows the activities of ALT, AST, ALP, and LDH in the serum of the rats fed a high-fat diet with orally administered cabbage-apple juice or fermented cabbage-apple juice for eight weeks with orally administered cabbage-apple juice or fermented cabbage-apple juice. The serum ALP activity did not show intergroup differences. The ALT, AST, and LDH activities decreased in the HFD-CA and HFD-FCA groups by approximately 14.4–19.7%, 7.3–16.9%, and 16.6–20.0%, respectively, as compared to the HFD group. However, a significant decrease when compared to HFD was displayed by HFD-FCA exclusively for ALT and AST and by both HFD-juice supplemented groups for LDH.

### 3.5. Serum Lipid Levels

Table 4 shows the changes in the serum lipid levels of the rats fed a high-fat diet with orally administered cabbage-apple juice or fermented cabbage-apple juice for eight weeks. The TG content in the serum decreased in the HFD-CA and HFD-FCA groups by 19.4% and 27.4%, respectively, compared to the HFD group. The TC content also decreased significantly in the HFD-CA and HFD-FCA groups by 19.1% and 26.5%, respectively. The LDL/VLDL-cholesterol content decreased in the HFD-CA and HFD-FCA groups by 13.6% and 23.2%, respectively, whereas the HDL-cholesterol content increased in the HFD-FCA group by 32.6%, as compared to the HFD group.

### 3.6. Serum Insulin, Glucose, Leptin, and Adiponectin Levels

Figure 3 shows the changes in serum insulin, glucose, leptin, and adiponectin levels. After the administration of cabbage-apple juice or fermented cabbage-apple juice, insulin levels decreased in the HFD-CA and HFD-FCA groups by 24.8% and 34.9%, respectively, compared to the HFD group. Glucose content in the serum was significantly decreased in the HFD-FCA group by 22.3% compared to the HFD group. Leptin levels showed a significant decrease in the HFD-CA and HFD-FCA groups by 19.2% and 29.4%, respectively, as compared to the HFD group, while the adiponectin levels only showed a significant increase in the HFD-FCA group, by 23.4%.

### 3.7. Hepatic Lipid Levels and Histopathological Changes

Figure 4 shows the changes in hepatic lipid levels as well as the morphological characteristics of the rats fed a high-fat diet with orally administered cabbage-apple juice or fermented cabbage-apple juice for eight weeks. The TG content in the liver decreased significantly in the HFD-CA and HFD-FCA groups by 33.9% and 44.4%, respectively, as compared to the HFD group. The TC content also decreased, by 11.2% and 20.8%, respectively, in the HFD-CA and HFD-FCA groups when compared to the HFD group. Notably, a significantly lower TC content was found in the HFD-FCA group as compared to the HFD group. When the livers of rats were extracted and stained by Red-O-Oil to examine lipid accumulation in liver tissue, the HFD group displayed numerous red-stained fat globules that clearly indicated lipid accumulation, whereas the HFD-juice supplemented groups (HFD-CA and HFD-FCA) showed fewer red-stained parts that indicated suppressed lipid accumulation. No differences in hepatic lipid accumulation were found between HFD-CA and HFD-FCA.

### 3.8. mRNA Expression of an Enzyme Related to Lipid Synthesis in the Liver

Figure 5 shows the effects on gene expression levels of the enzymes that are involved in hepatic lipid synthesis of the rats fed a high-fat diet with orally administered cabbage-apple juice or fermented cabbage-apple juice for eight weeks. The level of mRNA expression of ACC in the liver tissue was significantly lower in the HFD-CA and HFD-FCA groups than in the HFD group. On the contrary, the mRNA expression levels of FAS and G6PDH was significantly lower in the HFD-FCA group only. The malic enzyme gene expression levels in the liver did not show intergroup differences.

### 3.9. Epididymal Adipose Tissue TG Contents and Histopathological Changes

The TG content in the epididymal adipose tissue decreased in the HFD-juice supplemented groups (HFD-CA and HFD-FCA) by approximately 14.2–28.2% as compared to the HFD group (Figure 6). When the size of epididymal adipocytes was measured, HFD displayed a marked increase in size; however, a decrease in size in the HFD-juice supplemented groups (HFD-CA and HFD-FCA) when compared to the HFD group was shown in the HFD-juice supplemented groups (HFD-CA and HFD-FCA). HFD-FCA, in particular, showed a significantly reduced adipocyte size when compared to HFD-CA.

## 4. Discussion

The natural polyphenol compounds found in fruits and vegetables are known to exhibit anti-obesity effects by altering signal transduction in target cells, such as adipocytes, regulating gene expression, and enhancing free radical scavenging activity [12]. The polyphenol or flavonoid compounds abundant in apples include procyanidin, hydroxycinnamic acid and its derivatives, chlorogenic acid, caffeic acid, and epicatechin [9,10,11], while those that are abundant in cabbages include phenolic acids, flavonols, and anthocyanidines [19]. In addition, apples and cabbages are both rich in dietary fiber; notably, apples contain a high level of pectin among its dietary fiber that has been shown to act as a prebiotic in an in vivo study [42]. Cruciferous vegetables that belong to the family of cruciferous, such as cabbage, are rich in glucosinolates, carotenoids, and vitamin C, which play a major role in the modulation of lipid metabolism in vivo and in vitro [18]. Carotenoids have been reported to possess anti-obesity and anti-inflammatory abilities [43], and a hepatoprotective effect [22,23]. Such components are known to exert anti-cholesterol and anti-obesity activities [9,10,11,19]. In a previous study, when obese individuals were administered a 240–720 mg/kg of apple (fruits or juice) daily, body weight loss was observed; in experimental animals, a daily intake of 1–10 mg/kg of apples was shown to lead to body weight loss [12]. Consequently, we prepared a mixed juice containing equal amounts of apples and cabbages with known preventive effects on metabolic diseases that are attributed to obesity; next, by administering the juice to rats on a high-fat diet, we investigated the changes in body weight, liver and white fat pad weights, serum and hepatic lipid profiles, gene expression related to hepatic lipid metabolism, and adipocyte size. In our study, we have a limitation in vehicle control, since we administered saline and fiber into AIN-93M diet. The administration of saline and fiber cannot fully account for fiber and bioactive components in other groups.

The anti-obesity and hypolipidemic effects of vegetable and fruit juice fermented by LAB have been reported by several investigators [25,26,27,28,44]. LAB fermentation produces a variety of organic acids, short-chain fatty acids (SCFAs), amino acids, and secondary metabolite compounds [18,24,26,27,45,46]. Among the organic acids, SCFAs and amino acids showing anti-obesity properties in experimental animals are acetic acid [26], propionic acid [26,45], and ornithine [46]. In this study, *L. plantarum* EM fermentation showed an increase in dietary fiber, acetic acid, lactic acid, total organic acid, and total polyphenol contents, and a decrease in the crude fat and total free sugar contents of cabbage-apple juice.

Although soluble polyphenols are rapidly absorbed in the small intestine, most show a low absorption rate in the colon. The polyphenols contained in apples, in particular, are found in the form of aglycones and glucoside conjugates with low bioavailability [9,47]. Thus, fermented natural products have gained considerable attention because the fermentation of natural food ingredients with LAB has been shown to increase the bioactivity of nutrients through biotransformation or probiotic effects [9,48,49]. LAB converts the phenol compounds in fruits and vegetables to a more absorbable form in the human intestines, thereby maximizing the absorption rate and bioavailability [9,49]. Therefore, we conducted an experiment to compare non-fermented cabbage-apple juice and cabbage-apple juice fermented with kimchi-isolated *L. plantarum* EM [32] with respect to the anti-obesity effects and positive effects on lipid metabolism.

In this study, five-week old Sprague Dawley rats were fed a high-fat diet for eight weeks, which led to increased body weight, increased weights of liver and white fat pads, and increased levels of serum TG, TC, and LDL-cholesterol. The levels of TG and TC in liver tissue also increased, with increased expression levels of FSA, ACC, malic enzyme, and G6PDH genes that code for enzymes that are related to lipid synthesis, which confirmed body fat accumulation and dyslipidemia. These characteristics indicate that a high-fat diet induces obesity and hyperlipidemia, a result that coincides with previous studies on obesity [50]. In addition, the histopathological tests on liver tissue showed an increase in fat granules and lipid accumulation (hepatic steatosis). Epididymal fat pad size was also markedly increased. These phenomena resulted in significant increases in body weight and liver weight in experimental animals fed a high-fat diet [51]. The weight of the organs including the liver increased when high-fat diet caused unbalanced glycometabolism, inflow of excessively produced glucose, and abnormal RNA and DNA synthesis [52]. Obesity is caused by an increase in body fat, rather than in body weight, and an increase in the weight of adipose tissue leads to lipid accumulation, such that the higher the content, the higher the risk of metabolic disease. In particular, it is known that, despite equal body fat content, increased visceral fat rather than subcutaneous fat poses a health hazard; the higher the content of visceral fat, the higher the incidence of metabolic complications including changes in hypertension, dyslipidemia, and inflammatory cytokines, as well as in hyperinsulinemia resistance [53,54]. However, body weight loss in obesity improves obesity-associated diseases and metabolic disorders [55]. In this study, an inhibitory effect of abdominal obesity, as well as a reduced risk of metabolic disease, was observed, based on the decrease in not only body weight and the weights of the liver and white fat pads, but also in hepatic lipid accumulation and adipocyte size after the administration of cabbage-apple juice or fermented cabbage-apple juice. Such findings may suggest that obesity in mice can be prevented by non-absorbable procyanidins, a type of flavonoid found in apples [56], while apple-derived pectin attenuates metabolic endotoxemia in rats with diet-induced obesity, thereby reducing body weight and inhibiting body fat accumulation [42]. Furthermore, procyanidins, a component in apples consisting of various polymerized catechins, are known to suppress pancreatic lipase inhibitory activity and TG absorption [57]. The bioactive compounds, such as polyphenols or flavonoid, and the dietary fibers contribute to the decrease in body weight and body fat content.

Serum ALT and AST are distributed in liver tissue; as enzymes that are involved in amino acid biosynthesis, their activities are promoted upon damage to the liver due to drugs or stress, such that they are used as indicators of liver damage [58]. These enzymes show increased activities in obesity because the condition leads to lipid accumulation in liver tissue and the production of lipid peroxides that in turn produce reactive oxygen species, which together damage the liver [59]. Serum ALP activity increases with hyperlipidemia or related complications, hepatobiliary obstruction, and liver diseases. Advanced injury to hepatocytes leads to increased ALP activity and consequent disturbance to bile acid excretion in the liver and intestines, which is known to increase serum cholesterol levels [60]. In addition, LDH activity changes upon the disturbances to bile secretion that is caused by the onset of hypercholesterolemia or lipid accumulation in the liver and intestines [61]. In general, an input of excess TG or cholesterol to the liver as part of dietary intake is known to result in fatty liver and damage to hepatocytes, as the excess TG or cholesterol binds to lipid acceptor apoprotein to form lipoprotein that cannot be excreted. It is presumed that either juice may improve serum or liver lipid metabolism and delay injury to hepatocytes, thereby brining about positive effects on the recovery and maintenance of liver function, based on these results and the findings of this study demonstrating that the activities of AST, ALT, ALP, and LDH were increased by high-fat diet and reduced by administering cabbage-apple juice or fermented cabbage-apple juice. In an animal model with acetaminophen-induced liver damage, cabbage extract effectively lowered the activities of ALT and AST to exert an enhancing effect on liver protection and liver function [62]. In aerobic condition, lactic acid is increased to produce cellular energy via increasing LDH activity. The elevation of LDH activity is a pathological biomarker in cancer [63]. Therefore, the consumption of lactic acid should be carefully evaluated, since it might increase LDH activity because the fermented cabbage and apple juice inherently has higher lactate. Interestingly, one of clinical study demonstrated that short-term infusion of lactate did not alter metabolic rate and cytokine significantly [64]. Moreover, long-term exposure of lactate decreased LPS-inducible cytokine expression [64]. Therefore, dietary lactic acid might act differently when compared to endogenous lactic acid. However, further intensive studies are required to examine the potential net benefic and side effect in lactic acid consumption.

High-fat diet increases the incidence of atherosclerotic coronary artery disease and cardiovascular disease, while facilitating the induction of atherosclerosis and other complications [59]. However, the serum TG, TC, and LDL-cholesterol levels that increased in response to a high-fat diet were restored to healthy levels by administering cabbage-apple juice or fermented cabbage-apple juice since a reduction in serum lipid levels is known to reduce the risk of atherosclerotic cardiovascular disease [65]. The polyphenol compounds in fruits and vegetables play a beneficial role in preventing cardiovascular disease as they change the serum lipid levels [66]; furthermore, studies report that the higher the intake of polyphenol compounds, including flavonoids, the lower the mortality risk due to cardiovascular disease [67]. In a study that fed corn oil-loaded mice with a diet containing 60 mg of apple polyphenol, reduced TG absorption and serum TG content were observed [57]. Procyanidins, in particular, as a component of apple polyphenol, has been shown to exert anti-atherosclerosis and cholesterol-lowering effects in rats [68]. Cabbage leaf protein concentrate was also shown to have enhancing effects on serum lipid metabolism [69]. Moreover, cabbage extract, as well as the S-methyl-l-cysteine sulfoxide found in cabbages, were shown to inhibit hypercholesterolemia in hepatoma-bearing rats, which was attributed to a reduction in the serum cholesterol level and an increase in bile acid excretion in feces [21].

The liver is an organ that plays a crucial role in regulating the serum TG and TC levels by mediating the biosynthesis of TG and TC and their secretion to the circulatory system in lipoprotein forms. This renders TG and TC levels in the liver a key indicator of circulatory disease. The TG content in the liver depends on various interactions, such as the input, biosynthesis, oxidation, and release of fatty acids in VLDL form. High-fat diet causes excessive fatty acid input to the liver to induce lipid accumulation and a direct correlation between the accumulation of TG in the liver and insulin resistance has been reported [70]. The accumulation of TG and TC in the liver is known to increase liver weight. The high-fat diet applied in this study was shown to result in hepatic lipid accumulation that was based on increased TG, TC, and LDL/VLDL-cholesterol contents in liver tissue. Exposure to a high-fat diet was also shown to induce typical lesions in non-alcoholic fatty liver disease; upon histopathological observation, a myriad of isolated forms of lipid accumulation were found in the cytoplasm of hepatocytes. However, the TG, TC, and LDL/VLDL-cholesterol levels in liver tissue decreased when cabbage-apple juice or fermented cabbage-apple juice was administered, which implied that these juices might exert preventive effects on hepatic lipid accumulation. The formation of TG in the liver is based on synthesis mediated by several key enzymes such as ACC, FAS, and G6PDH; hence, gene expression levels for these lipid synthesis-related enzymes were also measured. Although high-fat diet increased the gene expression levels of ACC, FAS, malic enzyme, and G6PDH in liver tissue, administering cabbage-apple juice or fermented cabbage-apple juice led to a decrease in the expression of these genes. Thus, the results of reduced liver weight and TG content, as well as the inhibition of hepatic lipid accumulation by the administration of cabbage-apple juice or fermented cabbage-apple juice, seem to be the result of reduced gene expression levels of the lipid synthesis-related enzymes—malic enzyme, ACC, and FAS—and not to food intake.

The long-term intake of a high-fat diet induces elevated blood glucose levels and eventually leads to insulin resistance. In the human body, when the blood glucose level increases after a meal, the pancreas secrets insulin to lower the glucose level; however, in the case of excessive accumulation of body fat, hyperinsulinemia due to insulin resistance persists for a long time. Adipose tissue provides storage for excess energy, while it is also known to act as an endocrine organ that regulates body fat content and nutrient metabolism. An overabundance of adipose tissue and consequent dysfunction lead to a regulatory disorder of adipokine secretion that contributes to inflammatory reactions as well as changes in glucose and lipid metabolism, thereby resulting in obesity-associated metabolic diseases, including dyslipidemia, non-alcoholic fatty liver, insulin resistance, and type 2 diabetes [71,72]. Leptin is secreted from adipose tissue. It regulates appetite by stimulating the hypothalamus in the brain and glucose and lipid metabolism by increasing thermogenesis. The concentration of leptin is closely related to body fat content; in overweight or obese people, despite increased leptin levels, leptin resistance leads to the excess accumulation of TG in adipose tissue, liver, muscle, and pancreas, whereby insulin sensitivity and secretion are damaged [73]. In contrast, plasma adiponectin shows an inverse correlation with body fat content, such that its level decreases in obesity; moreover, with regard to insulin sensitivity, a positive correlation has been found [74]. Plasma adiponectin has also been reported to exert a positive effect on lipid metabolism, whereby plasma TG decreases but HDL-C increases, and the oxidation of fatty acids in the liver and muscle is facilitated along with lipoprotein lipase activity that decomposes VLDL to reduce the serum TG content [75,76]. A reduction in plasma adiponectin level has been shown to increase the risk of dyslipidemia and cardiovascular disease in experimental animals as well as humans [77]. In this study, although food intake did not take leptin function into account, the FGA-HFD group that was fed a high-fat diet and administered with fermented cabbage-apple juice showed a significant reduction in total fat content along with an increase in serum leptin levels, but a decrease in adiponectin levels as compared to the HFD group fed a high-fat diet only, suggesting that fermented cabbage-apple juice acted to reduce body fat content. The juice was also shown to enhance serum glucose and insulin levels in addition to adipokines, which suggested that it might improve obesity-induced metabolic diseases.

Fermented fruit-vegetable juice, as compared to non-fermented fruit-vegetable juice, has been reported to exhibit diverse health-promoting effects, such as enhancing the nutritional values, the bioavailability of phenolics, the contents and composition of secondary metabolite compounds, as well as antioxidant effects [9,26,27,44,45,46,48,49]. Such findings indicate that the enhanced lipid metabolism and anti-obesity effects are based on the compositional changes in natural polyphenol compounds [9,10,11], glucosinolates [18], and carotenoids [22,23,26,43] that are found in fruits and vegetables, the source materials of the juice, and dietary fiber [42], as well as organic acids [26,45], and amino acids [46] produced during LAB fermentation. Li et al. [26] reported that carrot juice that was fermented by *L. plantarum* NCU116 had greater acetic acid, propionic acid, lactic acid, β-carotene, and amino acid contents, and anti-diabetic, anti-oxidative, and lipid-lowering effects than those of the non-fermented carrot juice. Thus, when fermented cabbage-apple juice was compared with non-fermented cabbage-apple juice in this study, improvements in the compositional changes in serum lipids, gene expression regulation for enzymes engaged in lipid synthesis, and inhibitory effects on leptin appear to be the result of increased content or bioavailability of the functional ingredients in apples and cabbages after fermentation. Based on these findings, the different types of polyphenol compounds, dietary fibers, organic acids, and SCFAs contained in fermented cabbage-apple juice are presumed to have an effect in preventing obesity and improving lipid metabolism. Nevertheless, further studies are necessary for identifying the components and elucidating their effects on metabolic mechanisms.

## 5. Conclusions

Based on our findings, the intake of cabbage-apple juice or fermented cabbage-apple juice along with a high-fat diet appears to be effective in preventing various metabolic disorders that are caused by obesity, as the juice effectively regulates body weight and the weights of liver and white fat pads in rats with high-fat diet-induced obesity. Furthermore, it reduces the levels of serum leptin and insulin while increasing the level of adiponectin and altering gene expression for enzymes that are related to hepatic lipid metabolism to improve serum lipid levels. The anti-obesity effects and positive effects on lipid metabolism were shown to be more substantial in fermented cabbage-apple juice than non-fermented cabbage-apple juice. The cabbage-apple juice that was fermented with *L. plantarum* EM was shown to further enhance the beneficial effects of cabbage-apple juice on obesity-induced metabolic syndrome, at least under the conditions provided in this study.

## Figures and Tables

**Figure 1 nutrients-12-01135-f001:**
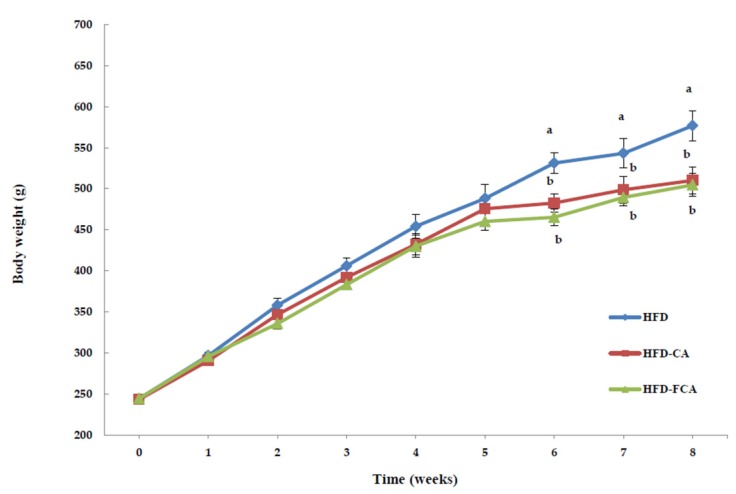
Body weight changes in rats fed experimental diets for 8 weeks. Body weight was measured weekly and presented as mean ± SE (*n* = 8). a,b; Bars with different letters are significantly different at *p* < 0.05 by Tukey’s test. Diet groups; HFD, high fat diet group: HFD-CA, HFD + cabbage-apple juice administration group: HFD-FCA, HFD + fermented cabbage-apple juice administration group.

**Figure 2 nutrients-12-01135-f002:**
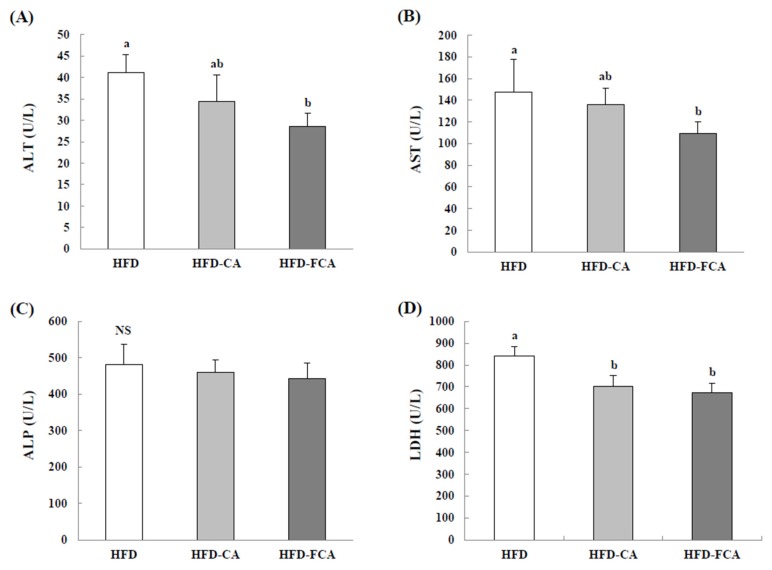
Serum alanine aminotransferase (ALT) (**A**), aspartate aminotransferase (AST) (**B**), alkaline phosphatase (ALP) (**C**), and lactate dehydrogenase (LDH) (**D**) activities in rats fed experimental diets for eight weeks. Values are mean ± SE (*n* = 8). a,b; Bars with different letters are significantly different at *p* < 0.05 by Tukey’s test. NS: No significance. Diet groups; HFD, high fat diet group: HFD-CA, HFD + cabbage-apple juice administration group: HFD-FCA, HFD + fermented cabbage-apple juice administration group.

**Figure 3 nutrients-12-01135-f003:**
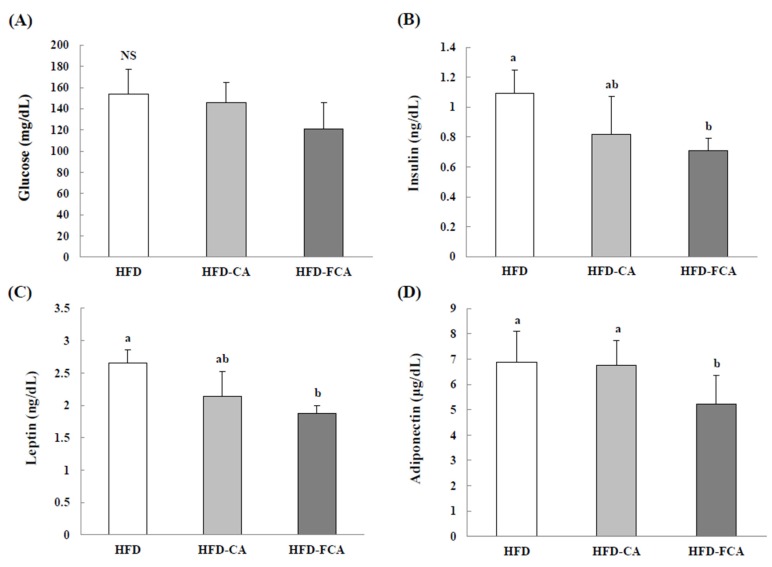
Glucose (**A**), insulin (**B**), leptin (**C**), and adiponectin (**D**) levels in serum of rats fed experimental diets for eight weeks. Values are mean ± SE (*n* = 8). a,b; Bars with different letters are significantly different at *p* < 0.05 by Tukey’s test. NS: No significance. Diet groups; HFD, high fat diet group: HFD-CA, HFD + cabbage-apple juice administration group: HFD-FCA, HFD + fermented cabbage-apple juice administration group.

**Figure 4 nutrients-12-01135-f004:**
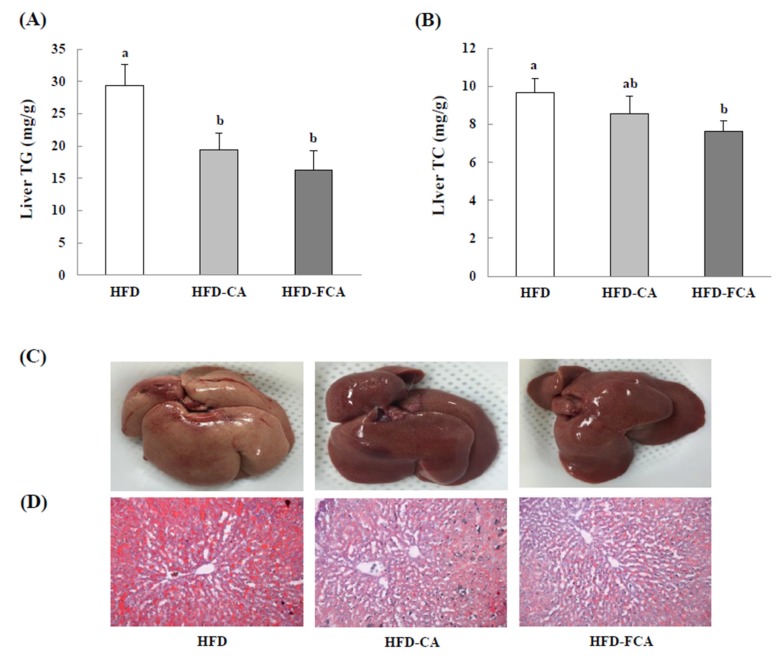
Hepatic triglyceride (**A**) and total cholesterol (**B**) levels, representative anatomical views (**C**), and histopathological analysis (**D**) in rats fed experimental diets for 8 weeks. All sections were stained with Oil Red O, ×100. Values are mean ± SE (*n* = 8 rats per group). a,b; Bars with different letters are significantly different at *p* < 0.05 by Tukey’s test. Diet groups; HFD, high fat diet group: HFD-CA, HFD + cabbage-apple juice administration group: HFD-FCA, HFD + fermented cabbage-apple juice administration group.

**Figure 5 nutrients-12-01135-f005:**
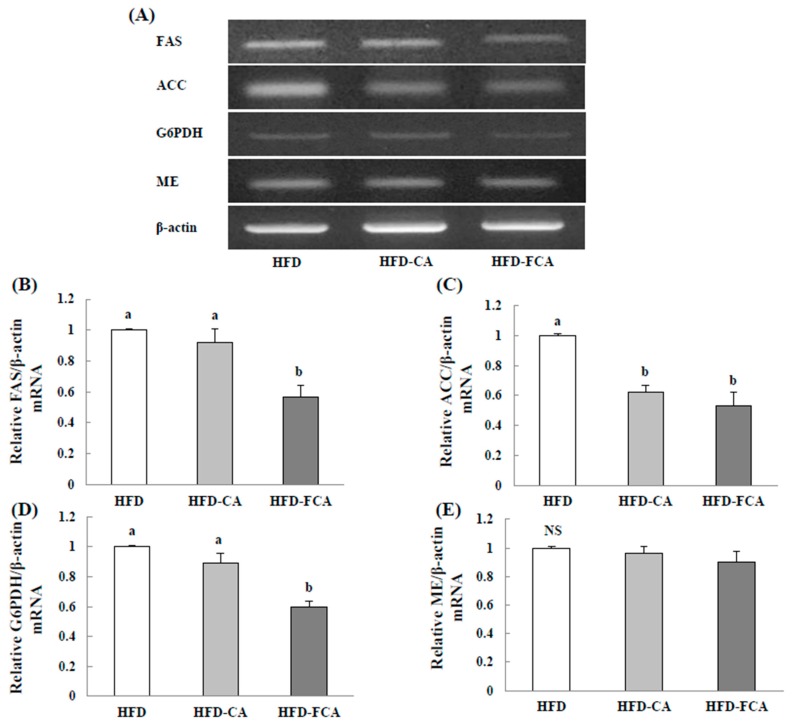
mRNA expression levels of enzymes related to lipid synthesis (**A**) in livers of rats fed experimental diets for eight weeks. The mRNA expression levels of FAS (**B**), ACC (**C**), G6PDH (**D**), and malic enzyme (ME) (**E**) were measured by RT-PCR. In the determination of mRNA levels, β-actin served as a loading control. Values are mean ± SE (*n* = 8 rats per group). a,b; Bars with different letters are significantly different at *p* < 0.05 by Tukey’s test. Diet groups; HFD, high fat diet group: HFD-CA, HFD + cabbage-apple juice administration group: HFD-FCA, HFD + fermented cabbage-apple juice administration group.

**Figure 6 nutrients-12-01135-f006:**
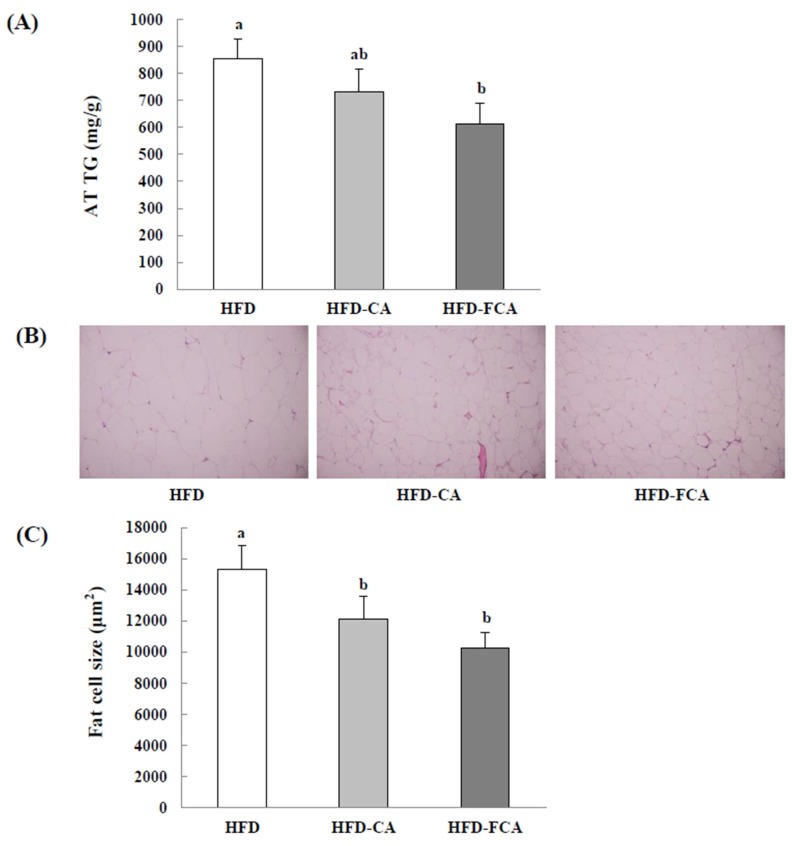
Epidydimal triglyceride content (**A**), representative findings (**B**), and adipocyte size (**C**) in rats fed experimental diets for 8 weeks. Epididymal fat tissues were visualized by hematoxylin and eosin staining. Adipocyte size was measured using a microscope and quantified using an image analyzer. Values are mean ± SE (*n* = 8 rats per group). a,b; Bars with different letters are significantly different at *p* < 0.05 by Tukey’s test. Diet groups; HFD, high fat diet group: HFD-CA, HFD + cabbage-apple juice administration group: HFD-FCA, HFD + fermented cabbage-apple juice administration group.

**Table 1 nutrients-12-01135-t001:** Changes in pH value, acidity, proximate composition, organic acid, free sugar, total polyphenol, and glucosinolates contents in non-fermented cabbage-apple juice and fermented cabbage-apple juice with *L. plantarium* EM.

	CA Juice	FCA Juice
pH	4.08 ± 0.02 ***	3.63 ± 0.03
Acidity (%)	1.26 ± 0.04 **	1.58 ± 0.04
Proximate composition (g/100 mL)		
Carbohydrate	10.10 ± 0.09	9.21 ± 0.07
Crude fat	3.72 ± 0.03 ***	1.52 ± 0.01
Crude protein	0.70 ± 0.02 **	0.61 ± 0.01
Moisture	89.45 ± 0.23 *	90.31 ± 0.45
Ash	0.62 ± 0.02	0.61 ± 0.01
Total dietary fiber	0.91 ± 0.01 ***	1.12 ± 0.03
Organic acid (g/100 mL)		
Citric acid	0.04 ± 0.00	ND
Malic acid	0.56 ± 0.01 ***	0.26 ± 0.00
Fumaric acid	ND	ND
Acetic acid	0.32 ± 0.00 ***	0.42 ± 0.00
Lactic acid	ND	1.45 ± 0.01
Total organic acid	1.03 ± 0.01 ***	2.13 ± 0.01
Free sugars (g/100 mL)		
Sucrose	1.44 ± 0.00 ***	1.30 ± 0.30
Glucose	3.22 ± 0.01 ***	2.17 ± 0.00
Xylose	0.05 ± 0.00	ND
Galactose	ND	0.07 ± 0.00
Fructose	5.07 ± 0.01 ***	4.86 ± 0.01
Sorbitol	0.20 ± 0.00 ***	0.20 ± 0.00
Total free sugar	9.98 ± 0.01 ***	8.59 ± 0.01
Total polyphenol (mg TAE/100 mL)	39.88 ± 2.68	42.36 ± 3.21
Total glucosinolates (mg/100 mL)	402.01 ± 20.32	365.55 ± 22.46

CA juice, non-fermented cabbage-apple juice; FCA juice, fermented cabbage-apple juice with *L. plantarum* EM. ND; not detected. Values are expressed as mean ± SE of experiments performed in triplicate. Significantly different between CA juice and FCA juice by Student’s *t*-test at * *p* < 0.05, ** *p* < 0.01, *** *p* < 0.001.

**Table 2 nutrients-12-01135-t002:** Changes in body weight gain and food intake of rats fed with experimental diets.

Group	Body Weight (g)				Food Intake (g/day)
	Initial Weight (g)	Final Weight (g)	Total Weight Gain (g)	Body Weight Gain (g/day)	
HFD	244.19 ± 2.29	576.90 ± 18.29 ^a^	332.64 ± 12.36 ^a^	5.94 ± 0.79 ^a^	22.72 ± 2.63 ^NS^
HFD-CA	243.19 ± 2.01	510.14 ± 15.89 ^b^	265.49 ± 11.13 ^b^	4.74 ± 0.47 ^b^	22.02 ± 1.87
HFD-FCA	244.25 ± 2.38	504.67 ± 14.33 ^b^	260.45 ± 9.88 ^b^	4.65 ± 0.38 ^b^	22.86 ± 2.25

Diet groups; HFD, high fat diet group: HFD-CA, HFD + cabbage-apple juice administration group: HFD-FCA, HFD + fermented cabbage-apple juice administration group. Values are mean ± SE (*n* = 8 rats per group). Values with different superscripts in the same column are significantly different (*p* < 0.05) among groups by Tukey’s test. NS: No significance.

**Table 3 nutrients-12-01135-t003:** Changes in the relative weight of the liver, mesenteric, epididymal, retroperitoneal, and total adipose tissues in rats fed experimental diets.

Group	Liver	White Fat Pads				
		Epididymal Fat Pads	Mesenteric Fat Pads	Retroperitoneal Fat Pads	Perinenal Fat Pads	Total White Fat Pads
	(g/100 g Body Weight)					
HFD	5.33 ± 0.58 ^a^	1.99 ± 0.33 ^a^	1.14 ± 0.57 ^a^	2.43 ± 0.62 ^a^	0.75 ± 0.16 ^NS^	6.31 ± 0.47 ^a^
HFD-CA	4.60 ± 0.30 ^b^	1.79 ± 0.42 ^b^	1.03 ± 0.18 ^ab^	2.36 ± 0.59 ^a^	0.71 ± 0.14	5.87 ± 0.65 ^b^
HFD-FCA	4.46 ± 0.28 ^b^	1.69 ± 0.62 ^b^	0.94 ± 0.20 ^b^	2.08 ± 0.40 ^b^	0.70 ± 0.07	5.32 ± 0.51 ^b^

Diet groups; HFD, high fat diet group: HFD-CA, HFD + cabbage-apple juice administration group: HFD-FCA, HFD + fermented cabbage-apple juice administration group. Values are mean ± SE (*n* = 8 rats per group). Values with different superscripts in the same column are significantly different (*p* < 0.05) among groups by Tukey’s test. NS: No significance.

**Table 4 nutrients-12-01135-t004:** Serum lipid profiles in rats fed experimental diets.

	HFD	HFD-CA	HFD-FCA
Triglyceride (mg/dL)	101.63 ± 11.97 ^a^	81.95 ± 8.16 ^b^	73.79 ± 8.22 ^b^
Total cholesterol (mg/dL)	123.51 ± 10.33 ^a^	99.88 ± 11.92 ^b^	90.60 ± 10.58 ^b^
LDL/VLDL cholesterol (mg/dL)	92.68 ± 9.94 ^a^	80.05 ± 7.02 ^ab^	71.19 ± 8.25 ^b^
HDL-cholesterol (mg/dL)	30.75 ± 5.97 ^b^	34.13 ± 4.12 ^b^	44.75 ± 4.53 ^a^

Diet groups; HFD, high fat diet group: HFD-CA, HFD + cabbage-apple juice administration group: HFD-FCA, HFD + fermented cabbage-apple juice administration group. Values are mean ± SE (*n* = 8 rats per group). Values with different superscripts in the same column are significantly different (*p* < 0.05) among groups by Tukey’s test. NS: No significance.

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
