# Peer review of "Effects of Cabbage-Apple Juice Fermented by Lactobacillus plantarum EM on Lipid Profile Improvement and Obesity Amelioration in Rats"

_nutrients, 2020, doi:10.3390/nu12041135_

Round 1
Reviewer 1 Report
The manuscript “Effects of Cabbage-apple Juice Fermented by Lactobacillus plantarum EM on Lipid Profile Improvement and Obesity Amelioration” reports the results of an investigation aimed at to exploring the role of a cabbage-apple juice on obesity and dyslipidemia induced by a high-fat diet in a rat model.
Authors have improved the manuscript considering comments from the past revision. However, some points still need to be addressed.
- Regarding the choice to use a mix of cabbage and apple, the explanations should be added somewhere in the text, with references if available.
- TAB 1: Authors improved the tables but there are still some components with potential effects that have not been considered (first of all glucosinolates from the cabbage)
- About the point "If authors wanted to discriminate whether fiber or bioactive compounds might have affected the body weight/lipid profile, a control group with just saline and fiber into AIN-93M diet should have been set" this reviewer was saying that it should have been considered in the study design. However, this point should be at least included as limitation of the study or as future perspective for future studies
- About the consideration that "Plantarum is a lactobacillus that produces lactic acid during its fermentative activity. How do authors explain that plasma LDH seems not to be affected by this increasing ingestion of lactic acid?", part of the explanation provided by authors (e.g. related to the correlation between lactate intake and LDH activity) can be added in the discussion as limitations of the study or as future perspectives for future investigations
Author Response
Reviewer 1
Comments and Suggestions for Authors
The manuscript “Effects of Cabbage-apple Juice Fermented by Lactobacillus plantarum EM on Lipid Profile Improvement and Obesity Amelioration” reports the results of an investigation aimed at to exploring the role of a cabbage-apple juice on obesity and dyslipidemia induced by a high-fat diet in a rat model.
Authors have improved the manuscript considering comments from the past revision. However, some points still need to be addressed.
- Regarding the choice to use a mix of cabbage and apple, the explanations should be added somewhere in the text, with references if available.
- Response: Authors appreciate the reviewer’s sincere and valued comment for this issue. Authors have revised the text accordingly. When we tested effect of various combinations of cabbage and apple juice to sensory panels, 1:1 ratio showed the highest score (data not shown). Thus, we selected 1:1 ratio in this study. . However, this point should be at least included as limitation of the study or as future perspective for future studies.
- TAB 1: Authors improved the tables but there are still some components with potential effects that have not been considered (first of all glucosinolates from the cabbage)
- Response: Authors have now added total glucosinolates data in Table 1.
- About the point "If authors wanted to discriminate whether fiber or bioactive compounds might have affected the body weight/lipid profile, a control group with just saline and fiber into AIN-93M diet should have been set" this reviewer was saying that it should have been considered in the study design. However, this point should be at least included as limitation of the study or as future perspective for future studies
- Response: As you suggested, we reflected and added your comments in our manuscript. [Line 382] In our study, we have a limitation in vehicle control since we administered saline and fiber into AIN-93M diet. Administration of saline and fiber cannot fully account fiber and bioactive components in other groups.
- About the consideration that "Plantarum is a lactobacillus that produces lactic acid during its fermentative activity. How do authors explain that plasma LDH seems not to be affected by this increasing ingestion of lactic acid?", part of the explanation provided by authors (e.g. related to the correlation between lactate intake and LDH activity) can be added in the discussion as limitations of the study or as future perspectives for future investigations
- Response: As you suggested, we reflected and added your comments in our manuscript. [Line 452] In aerobic condition, lactic acid is increased to produce cellular energy via increasing LDH activity. Elevation of LDH activity is a pathological biomarker in cancer (PMID: 26530363). Therefore, consumption of lactic acid should be evaluated carefully since it may increase LDH activity because the mixture of cabbage and apple juice inherently has higher lactate. Interestingly, one of clinical study demonstrated that short-term infusion of lactate did not alter metabolic rate and cytokine significantly (PMID: 30483253). Moreover, long-term exposure of lactate decreased LPS-inducible cytokine expression (PMID: 30483253). Therefore, dietary lactic acid may act differently compared to endogenous lactic acid. However, further intensive studies are required to examine the potential net benefic and side effect in lactic acid consumption.

Reviewer 2 Report
The manuscript "Effects of Cabbage-apple Juice Fermented by Lactobacillus plantarum EM on Lipid Profile Improvement and Obesity Amelioration in Rats" has been improved and in my opinion is now suitable for publication without any further changes.Author Response
We greatly appreciate the comments of reviewer.
Yours sincerely,
Jae-Joon Lee, Ph.D
Professor
Department of Food & Nutrition
Chosun University,
South Korea
Reviewer 3 Report
The authors responded appropriately to this reviewer's comments.
Author Response
We greatly appreciate the comments of reviewer.
Yours sincerely,
Jae-Joon Lee, Ph.D
Professor
Department of Food & Nutrition
Chosun University,
South Korea
Round 2
Reviewer 1 Report
The authors have improved the manuscript considering the comments from the past revision.
There is still a point in Table: authors should indicate which is the referring standard they used for the glucosinolate content in samples, as they did for polyphenol concentrations. More, it will be useful to express data as mg/mL being a liquid and not a solid and because being a unique referring standard, micromoles are not useful.
Also free sugars should be expressed g/100mL (as for the proximate composition) because numbers like 99,811.98 mg/L look strange
In this regard, which is the difference between total sugars and total free sugars? Are the former total carbs?
Author Response
Reviewer 1
There is still a point in Table: authors should indicate which is the referring standard they used for the glucosinolate content in samples, as they did for polyphenol concentrations. More, it will be useful to express data as mg/mL being a liquid and not a solid and because being a unique referring standard, micromoles are not useful.
Response: Authors appreciate the reviewer’s sincere and valued comment for this issue. We have expressed data in accordance with your comments and suggestions.
Also free sugars should be expressed g/100mL (as for the proximate composition) because numbers like 99,811.98 mg/L look strange
Response: We have expressed data, as advised. Amino acids were also expressed in g/100 mL
In this regard, which is the difference between total sugars and total free sugars? Are the former total carbs?
Response: Total sugars means carbohydrate. We have revised the word accordingly.
This manuscript is a resubmission of an earlier submission. The following is a list of the peer review reports and author responses from that submission.
Round 1
Reviewer 1 Report
Firstly, I would like to congratulate the authors for their work with fruit and vegetable (and not extracts) in relation to their possible biological effects. The manuscript is well written and with very interesting results. However, in its design, a control group is necessary to certify the results in obesity.
Moreover, in different regions of the world, each food has its composition, as well as after its processing. It is necessary to describe the nutritional composition of the foods used in the study, from the chemical composition (macronutrients and fibers) to the content of polyphenols.
In this sense, it is also necessary to describe better how you made sure that the juice was fermented.
Currently, in several studies using PCR, there is more than 1 internal control to guarantee the results. How do you explain the results found and justify the use of only 1 house-keeping? In fact, the PCR method must be better specified: was it with Syber? There is a figure of a gel (Figure 4). How was it done?
As well as, identify the sequences of the primers used to facilitate replication of the study by other interested authors.
Minor reviews:
- Line 38: the sentence is strange.
- Line 322: wouldn't it be "prebiotic"?
- Describe in the legend of the tables and figures the meaning of "a" and "b".
Author Response
Firstly, I would like to congratulate the authors for their work with fruit and vegetable (and not extracts) in relation to their possible biological effects. The manuscript is well written and with very interesting results. However, in its design, a control group is necessary to certify the results in obesity.
Response: The control group used in this study was selected with reference to various documents (Murphy EA, Velazquez KT, Herbert KM. Influence of high-fat diet on gut microbiota: a driving force for chronic disease risk. Curr Opin Clin Nutr Metab Care. 2015 Sep;18(5):515-20; Koh YM, Jang SW, Ahn TW. Anti-obesity effect of Yangkyuksanwha-tang in high-fat diet-induced obese mice. BMC Complement Altern Med. 2019 Sep 5;19(1):246; Tung YC 1, Chang WT , Li S , Wu JC , Badmeav V , Ho CT , Pan MH . Citrus peel extracts attenuated obesity and modulated gut microbiota in mice with high-fat diet-induced obesity. Food Funct. 2018 Jun 20;9(6):3363-3373.).
Moreover, in different regions of the world, each food has its composition, as well as after its processing. It is necessary to describe the nutritional composition of the foods used in the study, from the chemical composition (macronutrients and fibers) to the content of polyphenols.
Response: Authors appreciate the reviewer’s sincere and valued comment for this issue. Authors have now added chemical compositions data (macronutrients and fibers) in Table 1.
In this sense, it is also necessary to describe better how you made sure that the juice was fermented.
Response: Authors have revised the text accordingly.
Currently, in several studies using PCR, there is more than 1 internal control to guarantee the results. How do you explain the results found and justify the use of only 1 house-keeping? In fact, the PCR method must be better specified: was it with Syber? There is a figure of a gel (Figure 4). How was it done?
As well as, identify the sequences of the primers used to facilitate replication of the study by other interested authors.
Response: Authors have revised the text accordingly. However, in this study, reverse transcription-polymerase chain reaction was used instead of real-time PCR to measure gene expression.
Minor reviews:
- Line 38: the sentence is strange.
Response: Authors have revised the text accordingly.
- Line 322: wouldn't it be "prebiotic"?
Response: Authors have revised the word accordingly.
- Describe in the legend of the tables and figures the meaning of "a" and "b".
Response: Authors have described in the legend of the tables and figures the meaning of "a" and "b" accordingly.

Reviewer 2 Report
In this study, the authors described the effect of fermented and not fermented cabbage-apple juice in ameliorating the metabolic disorders caused by obesity. The study has a clinical relevance and give more insight into the role of probiotics and a fruit-vegetable mixture in the improvement of obesity and lipid profile. I found very interesting that fermented cabbage-apple diet and the cabbage-apple diet prevent various metabolic disorders caused by obesity and the fermented cabbage-apple diet enhances the anti-obesity effects of the cabbage-apple juice.The English writing is good, but the sentences in the abstract and introduction tend to be too long and sometimes there are too many punctuations that make the reading sometimes difficult and unclear. I wrote some examples in the following comments.The methods, results and discussion sections are good organized and well written. The experiments and statistic are well performed and the aim of the study is clear and meaningful.However, some comments needs to be addressed.
Line 15-19: The sentence is very long and difficult to read. I would suggest to the authors to shorten the sentence and to avoid writing a list of parameters. I would also suggest to write in the abstract the verbs in the active form and not in the passive form.
Here an example:
“In the HFD-fermented cabbage-15 apple juice-administered groups the following parameters decreased: body weight, liver and white fat pad weights, serum triglyceride 16 (TG), total cholesterol (TC), LDL-cholesterol, insulin, glucose and leptin levels, TG levels while HDL-C and adiponectin levels in serum increased compared with the HFD group. “
Line 21: please clarify FAS and ACC. Write the long name and then the short name in parenthesis.
Line 32: Please short the sentence as following: “An increase in fruit and vegetable intake has consistently been reported to reduce mortality due to cardiovascular disease and the risk of hypertension and stroke”
Line 34: Please change the sentence as following: “Fruits and vegetables are rich in potassium, folic acid, vitamins, dietary fiber, and phenol compounds. These compounds support the homeostasis regulation by decreasing oxidative stress, enhancing blood lipid metabolism, reducing blood pressure, and increasing insulin resistance.”
Line 37: Change “Such reports recommended” with “ It is recommended” with the references at the end of the sentence.
Line 40: Change the word “principal” with “ main”.
Line 42-44: Change the sentence as following: “Among the sugar components, 11-12% consists of oligosaccharides. In addition, the fruit contains a rich content of carotenoids, dietary fibers, vitamins, minerals, and antioxidant substances”.
Line 78: Please write the right acronyms generally recognized as safe (GRAS).
Line 79-81: Clarify the rest of the sentence “with both live and dead cells that show an outstanding ability to assimilate cholesterol and lower blood cholesterol”.
Line 111: How much juice was orally administrated to the rats? The authors should specify this information.
Line 181-194: It would be helpful to add an extra figure that show the weight change in the time (weeks).
Figure 4: Please explain every figure in the legend referring to the letters A,B,C,D,E.
Line 322: Did the author mean prebiotic or probiotic? If the sentence is referring to the pectin, the right terminology is prebiotic.
Line 332: The lower absorption rate “in the body” refers to the colon?
If so the authors should change the word “in the body” with “colon”.
Line 347-352: Please abbreviate the sentence.
Line 359: Add a point after reference 42 and start a new sentence.
Line 363-367: Change the sentence writing “Such findings may suggest”. The author didn’t demonstrate the statement, but they can hypothesize that obesity in mice can be prevented by non-absorbable procyanidins and by apple-derived pectin.
Line 368-370: Change the sentence as following: “The bioactive compounds such as polyphenols or flavonoid and the dietary fibers contribute to the decrease in body weight and body fat content”.
Line 460: Change the sentence as following: “Fermented fruit-vegetable juice, compared to non-fermented fruit-vegetable juice, has been reported to exhibit diverse health-promoting effects such as enhancing the bioavailability of phenolics, the contents and composition of polyphenol compounds, as well as antioxidant effects”.
Author Response
In this study, the authors described the effect of fermented and not fermented cabbage-apple juice in ameliorating the metabolic disorders caused by obesity. The study has a clinical relevance and give more insight into the role of probiotics and a fruit-vegetable mixture in the improvement of obesity and lipid profile. I found very interesting that fermented cabbage-apple diet and the cabbage-apple diet prevent various metabolic disorders caused by obesity and the fermented cabbage-apple diet enhances the anti-obesity effects of the cabbage-apple juice.The English writing is good, but the sentences in the abstract and introduction tend to be too long and sometimes there are too many punctuations that make the reading sometimes difficult and unclear. I wrote some examples in the following comments.The methods, results and discussion sections are good organized and well written. The experiments and statistic are well performed and the aim of the study is clear and meaningful.However, some comments needs to be addressed.
Line 15-19: The sentence is very long and difficult to read. I would suggest to the authors to shorten the sentence and to avoid writing a list of parameters. I would also suggest to write in the abstract the verbs in the active form and not in the passive form.
Here an example:
“In the HFD-fermented cabbage-15 apple juice-administered groups the following parameters decreased: body weight, liver and white fat pad weights, serum triglyceride 16 (TG), total cholesterol (TC), LDL-cholesterol, insulin, glucose and leptin levels, TG levels while HDL-C and adiponectin levels in serum increased compared with the HFD group. “
Response: Authors appreciate the reviewer’s sincere and valued comment for this issue. Authors have revised the text accordingly.
Line 21: please clarify FAS and ACC. Write the long name and then the short name in parenthesis.
Response: Authors have revised the text accordingly.
Line 32: Please short the sentence as following: “An increase in fruit and vegetable intake has consistently been reported to reduce mortality due to cardiovascular disease and the risk of hypertension and stroke”
Response: Authors have revised the text accordingly.
Line 34: Please change the sentence as following: “Fruits and vegetables are rich in potassium, folic acid, vitamins, dietary fiber, and phenol compounds. These compounds support the homeostasis regulation by decreasing oxidative stress, enhancing blood lipid metabolism, reducing blood pressure, and increasing insulin resistance.”
Response: Authors have revised the text accordingly.
Line 37: Change “Such reports recommended” with “ It is recommended” with the references at the end of the sentence.
Response: Authors have revised the text accordingly.
Line 40: Change the word “principal” with “ main”.
Response: Authors have revised the word accordingly.
Line 42-44: Change the sentence as following: “Among the sugar components, 11-12% consists of oligosaccharides. In addition, the fruit contains a rich content of carotenoids, dietary fibers, vitamins, minerals, and antioxidant substances”.
Response: Authors have revised the text accordingly.
Line 78: Please write the right acronyms generally recognized as safe (GRAS).
Response: Authors have made revision and improvement accordingly.
Line 79-81: Clarify the rest of the sentence “with both live and dead cells that show an outstanding ability to assimilate cholesterol and lower blood cholesterol”.
Response: Authors have revised the text accordingly.
Line 111: How much juice was orally administrated to the rats? The authors should specify this information.
Response: Authors have made revision and improvement accordingly.
Line 181-194: It would be helpful to add an extra figure that show the weight change in the time (weeks).
Response: Authors have now added an extra figure that show the weight change in the time (weeks) data in Figure 1to the manuscript.
Figure 4: Please explain every figure in the legend referring to the letters A,B,C,D,E.
Response: Authors have explained every figure in the legend referring to the letters A,B,C,D,E. to the manuscript.
Line 322: Did the author mean prebiotic or probiotic? If the sentence is referring to the pectin, the right terminology is prebiotic.
Response: Authors have changed the word accordingly.
Line 332: The lower absorption rate “in the body” refers to the colon?
If so the authors should change the word “in the body” with “colon”.
Response: Authors have changed the word accordingly.
Line 347-352: Please abbreviate the sentence.
Response: We have made revision and improvement accordingly.
Line 359: Add a point after reference 42 and start a new sentence.
Response: We have made revision and improvement accordingly.
Line 363-367: Change the sentence writing “Such findings may suggest”. The author didn’t demonstrate the statement, but they can hypothesize that obesity in mice can be prevented by non-absorbable procyanidins and by apple-derived pectin.
Response: We have made revision and improvement accordingly.
Line 368-370: Change the sentence as following: “The bioactive compounds such as polyphenols or flavonoid and the dietary fibers contribute to the decrease in body weight and body fat content”.
Response: We have made revision and improvement accordingly.
Line 460: Change the sentence as following: “Fermented fruit-vegetable juice, compared to non-fermented fruit-vegetable juice, has been reported to exhibit diverse health-promoting effects such as enhancing the bioavailability of phenolics, the contents and composition of polyphenol compounds, as well as antioxidant effects”.
Response: We have made revision and improvement accordingly.

Reviewer 3 Report
The manuscript “Effects of Cabbage-apple Juice Fermented by Lactobacillus plantarum EM on Lipid Profile Improvement and Obesity Amelioration” reports the results of an investigation aimed at to exploring the role of a cabbage-apple juice on obesity and dyslipidemia induced by a high-fat diet in a rat model.
I have some major concerns regarding the design of the study and the causal relationship between the results and conclusion of the study.
- Title is not reflecting the study design, at the end of the title a “in rats” should be added to let the reader know that it is not in humans.
- Why authors chose the mix of cabbage and apple? And why that cultivar of cabbage and that of apple?
- Concerning to the previous comment, the main limit of this study is the absence of the characterization of the juice, both the CA ad the FCA. So, actually, it is just a speculation of all the discussion saying that those compounds might be the ones having an effect. Furthermore, cabbage is very rich in glucosinolates and also carotenoids, which have not been mentioned. In my opinion, most of the effects might be done by soluble fiber, but this should be proofed. Moreover, in the fermentation process, the fibers are further chemically modified with the production of some metabolites, like SCFA, missing in this manuscript. Authors cannot proof what is in the beverages after the fermentation.
- Linked to the above comment, I do think that the comparisons in Table 1 are not adequate to fully understand the picture of results. My hypothesis is that rats who received the beverages had a higher intake of fiber compared to control group, which just received saline. So, the comparisons should be done within each group and before the treatment compared to the end of the treatment. Then, we can understand from this difference and from the comparisons of delta values among the groups whether the rats with beverages eat more/less than the control one. This is particularly important, as the amount of fats and the weight gain are very linked to them, and the amount of food is very worth to be considered.
- Related to the above comment, if authors wanted to discriminate whether fiber or bioactive compounds might have affected the body weight/lipid profile, a control group with just saline and fiber into AIN-93M diet should have been set.
- Can authors describe how did they record the amount of food eaten per day by rats? They usually jump into the jar of the food and urinate (and defecate) inside, so it is very tricky to weight the food. More, did they have unlimited access to food?
- L. Plantarum is a lactobacillus that produces lactic acid during its fermentative activity. How do authors explain that plasma LDH seems not to be affected by this increasing ingestion of lactic acid?
Author Response
The manuscript “Effects of Cabbage-apple Juice Fermented by Lactobacillus plantarum EM on Lipid Profile Improvement and Obesity Amelioration” reports the results of an investigation aimed at to exploring the role of a cabbage-apple juice on obesity and dyslipidemia induced by a high-fat diet in a rat model.
I have some major concerns regarding the design of the study and the causal relationship between the results and conclusion of the study.
- Title is not reflecting the study design, at the end of the title a “in rats” should be added to let the reader know that it is not in humans.
Response: Thank you for the reviewer’s comments for this title. Authors have changed the word accordingly.
- Why authors chose the mix of cabbage and apple? And why that cultivar of cabbage and that of apple?
Response: The reason for mixing apples and cabbage when making juice is that preliminary experiments showed that mixing cabbage and apples was the best sensory evaluation result. The cultivar of apple and cabbage was chosen because they are the most readily available cultivar on the market.
- Concerning to the previous comment, the main limit of this study is the absence of the characterization of the juice, both the CA ad the FCA. So, actually, it is just a speculation of all the discussion saying that those compounds might be the ones having an effect. Furthermore, cabbage is very rich in glucosinolates and also carotenoids, which have not been mentioned. In my opinion, most of the effects might be done by soluble fiber, but this should be proofed. Moreover, in the fermentation process, the fibers are further chemically modified with the production of some metabolites, like SCFA, missing in this manuscript. Authors cannot proof what is in the beverages after the fermentation.
Response: Authors appreciate the reviewer’s sincere and valued comment for this issue. Authors have now added chemical compositions (macronutrients and fibers) data in Table 1 to the manuscript. Authors have made revision and improvement accordingly.
- Linked to the above comment, I do think that the comparisons in Table 1 are not adequate to fully understand the picture of results. My hypothesis is that rats who received the beverages had a higher intake of fiber compared to control group, which just received saline. So, the comparisons should be done within each group and before the treatment compared to the end of the treatment. Then, we can understand from this difference and from the comparisons of delta values among the groups whether the rats with beverages eat more/less than the control one. This is particularly important, as the amount of fats and the weight gain are very linked to them, and the amount of food is very worth to be considered.
Response: Authors have mentioned the results (Table 1) of analyzing the nutrients, organic acid, and free sugar compositions, and total polyphenol contents between non-fermented cabbage-apple juice and fermented cabbage-apple juice cabbage-apple. The changes in body weight among the experimental groups are believed to be attributable to the differences in the chemical compositions and contents shown in Table 1 as well as dietary fiber. The differences in body weight before and after the experiment are presented in Table 2 as the body weight gain (g/day). There was no difference in food intake among the experimental groups, but the body weight gains decreased significantly in the juice-administrated groups compare to the control.
- Related to the above comment, if authors wanted to discriminate whether fiber or bioactive compounds might have affected the body weight/lipid profile, a control group with just saline and fiber into AIN-93M diet should have been set.
Response: Authors have made revision and improvement accordingly.
- Can authors describe how did they record the amount of food eaten per day by rats? They usually jump into the jar of the food and urinate (and defecate) inside, so it is very tricky to weight the food. More, did they have unlimited access to food?
Response: The experimental animals were bred one by one in a stainless steel cage, and food intake was checked daily. Urine and feces debris mixed with food in the feeder were removed using a sieve every day. Therefore, when calculated the food intake, the loss amount was also considered.
- L. Plantarum is a lactobacillus that produces lactic acid during its fermentative activity. How do authors explain that plasma LDH seems not to be affected by this increasing ingestion of lactic acid?
Response: Thank you for the reviewer’s comments for this issue. There was no study showing that LDH activity increased when animals were ingested with lactic acid in excess. AST, ALT, ALP and LDH activity in serum were measured indicators of liver damage. As a result, it was reported that the activity of these enzymes decreased after administration of LAB, thereby improving liver function. Increases in serum ALT, AST, and LDH activities in rats fed the HFD, in this study, tended to decrease by administration with non-fermented juice or fermented juice, and all rats in these groups showed normal activity ranges. These results indicate that non-fermented juice or fermented juice g does not cause hepatocellular toxicity. In this study, the correlation between lactate intake and LDH activity in serum was not studied. Therefore, there is a need to elucidate the detailed mechanisms of the effect of lactic acid intake on serum lactate concentration and LDH activity.
